Modulation of PI3K/AKT/mTOR and apoptosis pathway in colon cancer cells by the plant flavonoid fisetin

Alamoudi Amal
Alqarni Khlood
Khayyat Arwa Ishaq A.
Hussain Tajamul
Alamery Salman salamery@ksu.edu.sa
Biochemistry Department, College of Sciences, King Saud University , Riyadh , Saudi Arabia
Uversky Vladimir
Electronic publication date: 2025 Oct 27
Publication date: 2025
Volume: 13
Electronic Location ID: e20225
Received 2025 Apr 21; Accepted 2025 Sep 22
Copyright: ©2025 Alamoudi et al.
Copyright year: 2025
Copyright holder: Alamoudi et al.
License: This is an open access article distributed under the terms of the Creative Commons Attribution License, which permits unrestricted use, distribution, reproduction and adaptation in any medium and for any purpose provided that it is properly attributed. For attribution, the original author(s), title, publication source (PeerJ) and either DOI or URL of the article must be cited.
License URL: https://creativecommons.org/licenses/by/4.0/

Keywords: Fisetin, Flavonoid, Colon cancer, Gene expression, Cell proliferation, Apoptosis, PI3K/AKT/mTOR, Cellular pathways, BAX/BCL-2, Anticancer agent

Funding: King Saud University, Riyadh, Saudi Arabia ORF-2025-241 The authors received funding from the Ongoing Research Funding program, (ORF-2025-241), King Saud University, Riyadh, Saudi Arabia. The funders had no role in study design, data collection and analysis, decision to publish, or preparation of the manuscript.

==============================
Colorectal cancer (CRC) is a complex multifactorial disease caused by genetic and epigenetic changes playing a vital role in its development and progression. Chemotherapy remains a major option in the treatment of CRC. However, due to its unintended effects on normal tissue, research on identifying plant-based therapeutic agents as an alternative treatment modality has gained attention. Fisetin, a plant-derived flavonoid, has shown promising effects as an anticancer agent against several human cancers, including colon cancer. However, there is limited research focusing on studying the mechanism of action of fisetin. The PI3K/AKT/mTOR pathway as a key regulator of cancer cells has become a promising target for potential anti-cancer development. This study examined the anti-cancer effects of fisetin, emphasizing its effects on the PI3K/AKT/mTOR and apoptosis pathways in human colon cancer Caco-2 cells. The Caco-2 cells were treated with different concentrations (15, 30, 60, 90, or 120 µM) of fisetin for 12 or 24 h. Cell viability was evaluated using the MTT assay, while the expressions of PI3K/AKT/mTOR pathway genes and apoptosis genes, BAX and BCL-2, were analyzed by qRT-PCR. Fisetin markedly decreased the cell viability in a dose- and time-dependent manner. Fisetin down-regulated BCL-2, PI3K, mTOR, and NF-κB gene expression while up-regulating BAX gene expression. This suggested the inhibition of PI3K/AKT/mTOR pathway and induction of apoptosis. GeneMANIA and OncoDB further corroborated these results. These data demonstrate that the antiproliferative effects of fisetin were medicated through the modulation of PI3K/AKT/mTOR and apoptosis pathway. Thus, the study underscores fisetin’s potential as a cancer-preventative drug against cancer.

Introduction

Colorectal cancer (CRC) is one of the most prevalent types of gastrointestinal cancer that appears in the colon or rectum. It is the third most common cause of cancer-related deaths in men and the second in women worldwide, with an estimated one million people diagnosed with colorectal cancer each year, according to the World Health Organization (Bray et al., 2018; Suh et al., 2009).

CRC is a heterogeneous disease caused by genetic and epigenetic changes affecting different molecular pathways that eventually lead to the development of cancer. This usually happens due to the accumulation of genetic mutations or gene expression abnormalities. Therefore, it is essential to study the molecular basis of CRC to improve the diagnosis, prevention, and intervention (Markowitz & Bertagnolli, 2009). CRC pathogenesis has been shown to be associated with defects in multiple signaling pathways (Huang & Yang, 2022). Cellular activities like proliferation, apoptosis, and differentiation are mediated via intracellular signaling pathways, and mutations in genes regulating these pathways will transform normal cells to cancer cells (Huang & Yang, 2022). For example, genes regulating Wnt, KRAS/ MAPK, PI3K, TGF-β, and NF-κB pathways are frequently reported to be mutated or inactivated in human cancer (De Rosa et al., 2016; Hanahan & Weinberg, 2011; Sharma, Kelly & Jones, 2009).

Recently, PI3K/AKT/mTOR has been a major intracellular signaling pathway involved in regulating the cell cycle, cell proliferation, apoptosis, metabolism, and angiogenesis (Duan et al., 2018; Sun et al., 2021). This pathway is the most aberrantly activated in various human cancers because of its central role in critical intracellular signaling and important cellular processes (Khan et al., 2013; Yang et al., 2019). For instance, the PI3K gene in this pathway is frequently mutated in ∼20% of colorectal cancers (CRC) rendering it as a significant target for treatment (Samuels & Velculescu, 2004). Given its frequent activation in human cancer, the PI3K/AKT/mTOR pathway has become an attractive target for cancer therapy (Liu et al., 2018; Yap et al., 2008).

The therapeutic activity of nearly all investigated bioactive compounds on CRC has been related to PI3K/AKT/mTOR pathway suppression (Demir et al., 2024; He et al., 2021). For instance, quercetin, resveratrol, and curcumin exhibited cancer therapeutic effects by modulating genes related to the PI3K/AKT/mTOR pathway (Hasan et al., 2025; Hedayati et al., 2025; Johnson et al., 2009; Zoi et al., 2024). Thus, PI3K/AKT/mTOR pathway is still the most widely promising target for potential anti-cancer therapies. many small-molecule inhibitors based on this pathway have been developed (Cheng et al., 2005). For example, a novel anticancer drug, quinazolinone-chalcone derivative, is reported to interfere with the PI3K/AKT/mTOR signaling cascade in colon cancer HCT-116 cells (Wani et al., 2016). Likewise, another anticancer drug, cryptotanshinone (CPT), exerts its effects by inhibiting the PI3K/AKT/mTOR pathway in colon cancer (Zhang et al., 2018).

Plant-derived natural compounds have shown promising effects in cancer therapy. Importantly, these natural agents have proven to be safe for the normal tissues. Fisetin flavonoid, commonly found in fruits and vegetables such as strawberries, persimmons, cucumbers, apples, and onions, possesses anti-inflammatory, antioxidant, and anticancer effects. Fisetin has gained extensive attention for its remarkable biological activities, such as anticancer, antidiabetic, anti-inflammatory, antioxidant, and neuroprotective effects (Zhong et al., 2022). It is shown to inhibit cell proliferation, migration, and invasion in several human cancers through the induction of apoptosis (Haddad et al., 2006; Pal et al., 2013). In various cancer types, fisetin has been shown to suppress cell proliferation, migration, and invasion and to trigger apoptosis, such as in colon cancer (Chen et al., 2015a), lung cancer (Kang, Piao & Hyun, 2015), nasopharyngeal carcinoma (Li, Zeng & Shen, 2014a), prostate cancer (Chien et al., 2010), bladder cancer (Li et al., 2014b), and cervical carcinoma (Chou et al., 2013). Previous studies have provided evidence that fisetin might interfere with several signaling pathways that control cell survival, growth, and proliferation (Syed et al., 2016). Specifically, fisetin was shown to have anti-cancer activity by modulating PI3K/AKT/mTOR pathway in various types of cancer (Farooqi et al., 2021; Rychahou et al., 2006; Xiao et al., 2021; Zhang & Jia, 2016). In addition, fisetin down-regulated BCL-2 expression and induced apoptosis in lung cancer (Kang, Piao & Hyun, 2015).

Cheng et al. (2020) demonstrated that the inhibition of the PI3K/AKT/mTOR pathway results the inhibition of proliferation and induction of apoptosis of CRC cells after treatment with naringin. Furthermore, it suggested that the prevention and treatment of CRC through the PI3K/AKT/mTOR pathway is a feasible direction for future research (Cheng et al., 2020). Moreover, the apoptosis was induced via a mutant of PI3K/AKT/mTOR signaling cascade in colon cancer HCT-116 cells (Wani et al., 2016). Therefore, selective targeting of the PI3K/AKT/mTOR pathway may provide effective treatment of colorectal cancer (Rychahou et al., 2006). This promoted an investigation into the effects of fisetin on PI3K/AKT/mTOR and apoptosis pathway on colon cancer cell Caco-2, highlighting its significance as a target in CRC treatment.

Above all, fisetin has been extensively studied in the colon cancer cell lines HCT-116 and HT-29 but not in the Caco-2 cell line (Li et al., 2022; Suh et al., 2009; Syed et al., 2016). Therefore, we believe that ongoing in vitro studies focusing on blocking the PI3K/AKT/mTOR /NF-kB pathways would be interesting and provide promising results that may enhance the efficiency of anti-cancer therapeutics. Given that the inhibitory effects of fisetin on PI3K/AKT/mTOR pathway in colon cancer have not been fully investigated. In the present study, we studied the effects of fisetin on PI3K, AKT, mTOR, and apoptotic genes in Caco-2 cells.

Materials & Methods

Cell culture

The human adenocarcinoma colon cancer Caco-2 cells from the American Type Culture Collection (HTB-37 –ATCC, USA), derived from a human colorectal adenocarcinoma isolated from a 72-year-old male with colorectal cancer, were cultured in Dulbecco’s Modified Eagle Medium (DMEM) (Gibco®, Waltham, MA, USA), containing 10% Fetal Bovine Serum (FBS) (Gibco®, Waltham, MA, USA), and 1% penicillin-streptomycin antibiotic (UFC Biotech®, Amhurst, MA, USA). Cells were incubated in a humidified atmosphere (BINDER®, Tuttlingen, Germany) at 5% CO2 and 37 °C. Fisetin was dissolved in DMSO at 10 mM concentration (stock) and further diluted in DMEM to obtain desired concentrations.

MTT assay

The antiproliferative effects of fisetin were assessed using the MTT assay. CaCo-2 cells were seeded in a 96-well plate at a density of 103 cells/well in 200 µl complete media and incubated for 24 h at 37 °C and 5% CO2. After 24 h of incubation of cells, the medium was replaced by a fresh medium. At 80–90% confluency, cells were treated with 15, 30, 60, 90, or 120 μM concentrations of fisetin. Control cells were treated with vehicle (DMSO) only. Plates were kept on a shaker for 5 min for the complete dissolution of fisetin in the media and incubated for 12 or 24 h. Cell viability was assessed by using a commercially available Vybrant MTT cell proliferation Assay kit (V-13154, Molecular Probes®, Eugene, OR, USA) according to the manufacturer’s instructions. Ten µl of MTT reagent (5 mg/mL in PBS) was added to each well and further incubated for 4 h at 37 °C. Culture media containing MTT was discarded, and 100 µl of SDS-HCl solution (10 ml of 0.01 M HCL/1 gm SDS in sterile distilled water) was added to each well, and the plate was left in a humidified chamber overnight. The absorbance of the color was measured at 405 nm using BioTek ELx800 absorbance microplate reader (Thermo Fisher Scientific®, Waltham, MA, USA). The optical density (OD) value was subjected to sort out the percentage of cell viability by using the following formula:

Cell viability (%) = (OD value of experimental samples / OD value of experimental control sample) X 100.

RNA extraction and cDNA synthesis

Caco-2 cells were grown to 80% confluency in 5 ml of complete DMEM culture media as detailed above and treated with 60 µM, 90 µM and 120 µM for 12 or 24 h. Untreated cells (control) were treated with 1% DMSO. Thereafter, cells were collected, and total RNA was extracted using the RNeasy plus mini kit (Qiagen®, Hilden, Germany). The purity and concentration of RNA were determined using a Nanodrop ND-1000 spectrophotometer (Thermo Fisher Scientific, Epsom®, Waltham, MA, USA).

The cDNA was synthesized from 1 µg of total RNA by reverse transcription reaction using the SuperScript III First-Strand Synthesis System for RT-PCR Kit (Cat No. 18080-051) (Thermo Fisher Scientific®, Waltham, MA, USA) according to the manufacturer’s protocol instruction in BioRad My Cycler thermal cycler (BioRad®, Hercules, CA, USA).

Measurement of mRNA level

The mRNA level of BCL-2, BAX, mTOR, AKT, PI3K, and NF-κB genes was measured by qRT-PCR using the QuantiTect SYBR Green PCR kit from Qiagen (Hilden, Germany) according to the manufacturer’s instructions. We used the qPCR system SALAN Real-Time PCR Detection System® (Korea). Two microliters of the 10 µl synthesized cDNA from each sample were used in the qPCR reaction. The sequences of the primers of BCL-2, BAX, mTOR, AKT, PI3K, NF-κB and GAPDH are listed in Table 1. Each sample of cDNA was analyzed in duplicate for all the studied genes.

Table 1 Sequences of primers used in gene expression.

Genes	Forward primer	Reverse primer	
GAPDH	GTCTCCTCTGACTTCAACAGCG	ACCACCCTGTTGCTGTAGCCAA	
BCL-2	AATGGGCAGCCGTTAGGAAA	GCGCCCAATACGACCAAATC	
BAX	GGCCCAATACGACCAAATC	GGAAAAAGACCTCTCGGGGG	
AKT	TGGACTACCTGCACTCGGAGAA	GTGCCGCAAAAGGTCTTCATGG	
mOTR	CAAGAACTCGCTGATCCAAATG	GCTGTACGTTCCTTCTCCTTC	
PI3K	GAAGCACCTGAATAGGCAAGTCG	GAGCATCCATGAAATCTGGTCGC	
NF-κB	GCAGCACTACTTCTTGACCACC	TCTGCTCCTGAGCATTGACGTC	

Validation of gene expression results

The expression patterns of BCL-2, BAX, mTOR, AKT, PI3K, and NF-κB genes in Caco-2 cells were compared with The expression data in colon adenocarcinoma (COAD) tissues using the OncoDB portal https://oncodb.org/index.html (Tang et al., 2024) (accessed on 15 Jan 2025).

Co-expression network analysis

To examine the interaction network of potential genes involved in the pathway, we analyzed mTOR, AKT, PI3K, and NF-κB genes for physical interaction and co-expression using the GeneMANIA tool https://genemania.org/.

Statistical analysis of the results

The statistical software GraphPad Prism V.9 was used to perform statistical analysis. The relative gene expression was calculated using the comparative threshold cycle method (2−ΔΔCt) after normalizing to the values of the GAPDH housekeeping gene and relative control samples. Two-way ANOVA was used to analyze the associations between the group differences. A p-value of less than 0.05 was considered statistically significant.

Results

Anti-proliferative effects of fisetin on Caco-2 cells

The MTT assay was used to study the effects of fisetin treatment on Caco-2 cell viability and proliferation using indicated doses and time durations of 12 and 24 h (Fig. 1). The treatment of Caco-2 cells with fisetin showed cytotoxic activity by inhibiting cell proliferation when compared to untreated cells. Fisetin at 60, 90, and 120 µM doses significantly reduced the cell proliferation compared to the untreated control at both 12- and 24-hour times. Fisetin at 60, 90, and 120 µM doses showed dose-dependent antiproliferative effects at 12 and uniform effects at 24 h time durations, while at 15 and 30 µM concentrations, it had no significant effects compared to the control (Fig. 1). The optimum concentrations that provided maximum growth inhibition were 60 µM and 90 µM, respectively, as shown in Fig. 1. Moreover, fisetin treatment for 12 h exerted superior antiproliferative effects compared to 24 h.

Figure 1 Fisetin inhibits the proliferation of Caco2 cancer cells.

Cells were treated with indicated concentrations of fisetin at 12 or 24 h. Compared to control, fisetin at 60, 90 and 120 µM significantly and dose dependently reduced cell viability (p < 0.01, <0.001, <0.0001 respectively) after 12 h treatment. Similarly, fisetin treatment for 24 h at 60, 90 and 120 µM doses significantly reduced cell viability (p < 0.01, <0.01, <0.05 respectively) compared to control. Results are presented as mean ± SE (Standard error) of three independent experiments. A p-value < 0.05 was considered statistically significant. ns: non-significant. Asterisks indicate level of statistical significance: Significant *p ≤ 0.05, Very significant **p ≤ 0.01, ***Very significant p ≤ 0.001.

Fisetin downmodulates PI3K, mTOR and NF-κB gene expression

The PI3K, AKT, mTOR and NF-κB gene expression levels were measured by qRT-PCR after 12 and 24 h of fisetin treatment. The gene expression of PI3K, AKT, mTOR and NF-κB was significantly decreased in Caco-2 cells treated with 60 and 90 µM of fisetin compared to those in control at both the studied time durations (Fig. 2). Relatively, the fold change of studied genes was significantly lower after 24 h than after 12 h of fisetin treatment, indicating a more pronounced effect with longer fisetin treatment. No significant variations in the AKT mRNA levels were noted between the fisetin treated and control cells at both the studied time points. The effect of fisetin was more significant on PI3K and NF-κB gene expressions than on mTOR.

Figure 2 Effect of fisetin on PI3K, AKT, mTOR and NF-κB gene expression in colon cancer cells (Caco-2).

Fisetin at 60, 90 and 120 µM concentrations significantly decreased PI3K mRNA after 12 h (p < 0.001) or after 24 h (p < 0.0001), significantly decreased mTOR mRNA levels after 12 h (p < 0.001) or after 24 h (p < 0.0001) and NF-κB mRNA levels after 12 h (p < 0.001) and after 24 h (p < 0.0001) compared to control. No significant variations in mRNA levels of AKT were observed after both 12 h and 24 h fisetin treatments. Results are presented as mean ± SE (Standard error) of three independent experiments. A p-value <0.05 was considered statistically significant. ns: non-significant. Asterisks indicate level of statistical significance: Significant *p ≤ 0.05, Very significant **p ≤ 0.01, ***Very significant p ≤ 0.001.

Expression data from the oncoDB database indicate that the expression of PI3k is higher in colon cancer (n = 308) compared to normal tissues (n = 41), whereas the expression of AKT, mTOR and NF-κB genes showed lower expression (Fig. 3). But PI3K was down-regulated by fisetin in Caco-2 cells. This implies that fisetin may target the PI3K gene in Caco-2 cells.

Figure 3 Gene expression levels of the PI3k, AKT, mTOR and NF-κB genes in colon cancer compared to normal tissue obtained from the OncoDB database.

We further investigate the relationship between PI3K and AKT, mTOR and NF-κB genes utilizing the GeneMANIA program. It showed physical interactions (77.6%) and co-expression (8%) of PI3K with AKT, mTOR and NF-κB genes (Fig. 4). These genes are involved in cell proliferation and survival. Hence, these genes were tested for gene expression with and without treatments. These data are concurrent with our findings, where these genes were found to be concomitantly regulated by fisetin and support their involvement in the colon cancer etiology.

Figure 4 Network interaction for the PI3K, AKT, mTOR and NF-κB genes using GeneMANIA program.

Genes show physical interactions (77.6%) and co-expression (8%) with each other. These genes interacted with each other forming complexes or activation cascades in implicated pathways related to cell proliferation and survival.

Fisetin modulates BAX and BCL-2 genes expression

The data on the expression levels of BCL-2 and BAX were measured after 12 and 24 h of fisetin treatment as shown in Fig. 5. Fisetin significantly modulated the expression of both genes (BCL-2 and BAX) in Caco-2 cells, where their expression levels were significantly altered in a time- and dose-dependent manner when compared to untreated cells (Fig. 5). The mRNA level of the proapoptotic BAX gene expression gene was significantly up-regulated. While the mRNA level of the antiapoptotic BCL-2 gene was significantly down-regulated after 12 and 24 h of fisetin treatments compared to their respective levels in the control. In addition, the BAX/ BCL-2 gene expression ratio increased in response to fisetin and was higher at 24 h of treatment. Expression data from the oncoDB database indicate that the expression of BAX is up-regulated in colon cancer (n = 308) compared to normal tissues (n = 41), whereas the expression of Bcl-2 is down-regulated (Fig. 6). These results may clearly indicate that fisetin plays a role in the induction of apoptosis by inhibiting BCL-2 and activating BAX.

Figure 5 Effect of fisetin on BAX and BCL-2 gene expression in colon cancer cells (Caco-2).

Fisetin at 60, 90 and 120 µM concentrations significantly increased BAX after 12 h (p < 0.001) and 24 h (p < 0.0001) treatment and significantly decreased BCL-2 expression after 12 h (p < 0.001) and 24 h treatments compared to control. Results are presented as mean ± SE (Standard Error) of three independent experiments. A p-value <0.05 was considered statistically significant. ns: non-significant. Asterisks indicate level of statistical significance: Significant *p ≤ 0.05, Very significant **p ≤ 0.01, ***Very significant p ≤ 0.001.

Figure 6 Gene expression levels of the BCL-2 and BAX genes in colon cancer compared to normal tissue obtained from the OncoDB database.

Discussion

Fisetin is a natural plant-derived compound that has gained extensive attention due to its remarkable anticancer effects through modulation of multiple genes and signaling pathways. Previous studies have provided evidence that fisetin might interfere with signaling pathways that control cell survival, growth, proliferation, and apoptosis (Youns & Abdel Halim Hegazy, 2017). These include the NF-κB, MAPK, Wnt, PI3K/AKT/mTOR, and apoptosis pathways (Syed et al., 2016; Wang et al., 2022).This study was aimed to investigate the anti- cancer properties of fisetin in colon cancer Caco-2 cells by studying its effect on cell proliferation and the expression of cell survival pathway genes, including PI3K, AKT, and mTOR, as well as BCL-2 and BAX, which are involved in the apoptosis signaling pathways.

In the present study, the cell viability assay revealed a gradual decrease in the number of live cells with an increase of fisetin dose, demonstrating the inhibitory effects of fisetin on the cell proliferation of Caco-2 cells when compared to untreated. fisetin treatment for 12 h was enough to achieve significant cell growth inhibition. fisetin at 60, 90, and 120 µM concentrations for 12 h revealed a dose-dependent and robust cell growth inhibition in Caco-2 cells. These findings suggest the ability of fisetin to regulate cell proliferation via modulation of cell signaling pathways.

Our study findings are consistent with previous studies where fisetin was shown to inhibit cell proliferation in different cell types (Afroze et al., 2022; Lim & Park, 2009; Youns & Abdel Halim Hegazy, 2017). For example, a study done by Chien et al. (2010) demonstrated the effect of fisetin on the viability of HCT-116 and HT29 colon cancer cells after cells were treated with fisetin up to 240 µM for 24, 48, 72, and 96 h. However, the study showed that HT29 cells were more sensitive to fisetin as median inhibition concentration values were lower than that of HCT-116 cells.

In order to investigate the effect of fisetin on apoptotic genes such as BCL-2 (antiapoptotic gene) and BAX (proapoptotic gene), qPCR was carried out to assess the expression of BAX and BCL-2 in Caco-2 cells after treatment with fisetin. Results showed a significant increase in the BAX gene, whereas a decrease in BCL-2 protein was noted in response to increasing concentrations of fisetin. Furthermore, the BAX/ BCL-2 was significantly increased. These data demonstrate the proapoptotic nature of fisetin.

These findings are in line with the previous studies where fisetin was shown to modulate BCL-2 and BAX genes in triggering apoptosis in human osteosarcoma cells (Li et al., 2015). Similarly, fisetin suppressed the protein levels of antiapoptotic Bcl-XL and BCL-2 and increased proapoptotic BAK and BIM in HCT-116 colon cancer cells (Lim & Park, 2009). It has been found that fisetin has the potential to suppress cell proliferation of various cancer cells by inhibiting cell growth, inducing apoptosis, and halting the cell cycle (Qaed et al., 2023). Activation of BCL-2 expression in various cancers suggested the essential role of BCL-2 in apoptosis pathways (Lindsay, Esposti & Gilmore, 2011).

To unveil the underlying mechanisms of the anti-cancer effect of fisetin, a cell survival signalling pathway, PI3K/AKT/mTOR, was examined by measuring gene expression of PI3K, AKT, mTOR and NF-κB. The results demonstrated that treatment of Caco-2 cells with fisetin significantly reduced the expression of PI3K, mTOR, and NF-κB. But there were no significant changes in the AKT at the RNA level. This is because AKT is regulated at the protein level rather than the mRNA level. Fisetin decreases phosphorylation of AKT at the protein level as reported by previous studies (Khan et al., 2011; Syed et al., 2014). Furthermore, AKT was down-regulated at the protein expression level by curcumin (Jiang et al., 2014). Hence, our results suggest that fisetin may reduce AKT gene expression at the protein level through phosphorylation events, thereby inhibiting the proliferation of Caco-2 cells and inducing apoptosis.

The PI3K gene has been recognized as a significant oncogene in numerous cancers (Khan et al., 2013; Yap et al., 2008). Inhibition of PI3K can result in both decreased cellular proliferation and increased cellular death (Hennessy et al., 2005). Therefore, we found that PI3K gene expression decreased after fisetin treatment in Caco-2 cells, leading to the inhibition of cell proliferation. Previous studies also suggested that fisetin has antimetastatic potential linked to the inactivation of the PI3K/AKT in prostate cancer cells (Adhami et al., 2012). Furthermore, the fisetin treatment of human Non-small lung cancer cells (NSCLC) caused dual inhibition of PI3K/AKT/mTOR (Khan et al., 2012).

Fisetin had a potential inhibitory effect on PI3K and related genes such as AKT and mTOR, leading to reduced cell proliferation and inducing apoptosis. There is evidence that down regulated PI3K by fisetin also has an inhibitory effect on other genes such as AKT and BCL-2, suggesting the dual role of fisetin in modulating PI3K/AKT/mTOR and apoptosis pathway (Zhang & Jia, 2016). Similarly, the anti-cancer function of quercetin is mediated by targeting this pathway (Hasan et al., 2025).

NF-κB transcription factor is key regulators of innate and adaptive immune responses, inflammation, and cell survival (Sakamoto & Maeda, 2010). The NF-κB pathway has been implicated in various steps of cancer development, including initiation, proliferation, metastasis, and resistance to therapy (Vaiopoulos, Athanasoula & Papavassiliou, 2013). In this present study the NF-κB level was found to be significantly decreased by fisetin in Caco-2 cells. Consistently, other studies also found that fisetin down-regulated NF-κB in HT29 cells and pancreatic cancer cells (Murtaza et al., 2009; Suh et al., 2009). Hence, we supposed that the antiproliferative effects of fisetin were mediated by its ability to inhibit NF-κB expression as a result of inhibition of PI3K/AKT/mTOR pathway.

Figure 7 illustrates the effects of fisetin on PI3K/AKT/mTOR pathway. Fisetin modulates the pathways directly by the key genes in the pathway or indirectly by various upstream and downstream regulators such as KRAS and PTEN (Tong & Pelling, 2013). KRAS functions upstream of the pathway to interact and activate PI3K. Fisetin caused inhibition of this pathway by downregulating KRAS, leading to inhibition of cell proliferation and induction of apoptosis (Appari et al., 2014). In addition, PTEN negatively regulates this pathway. Previous studies reported that fisetin increased expression of PTEN resulting in the inactivation of PI3K and the reduction of phosphorylation of AKT and mTOR (Lall, Adhami & Mukhtar, 2016; Li et al., 2018; Pal et al., 2015).

Figure 7 Illustration the effect of fisetin on PI3K/AKT and mTOR pathway.

Activation is shown with arrowhead lines. Inhibition is indicated with roundhead lines. Dashed lines indicate indirect activation. Red lightning bolt indicate Fisetin’s target.

AKT acts as a key downstream signaling molecule for PI3K (Kizilboga et al., 2019). Activation of AKT by PI3K results in phosphorylation of a variety of downstream protein substrates, including mTOR , BAD, and NK-κB (Ghareghomi et al., 2021). NF-κB interplay with this pathway and plays an important role in the regulation of cancer, inflammation, and apoptosis (Ghoneum & Said, 2019). Activation of NF-κB by PI3K, AKT, and mTOR induces the expression of its target gene. mTOR is positively activated by AKT (Bjornsti & Houghton, 2004). It is reported that fisetin physically binds and inhibits the activity of mTOR (Khan et al., 2011; Syed et al., 2013). This also led to inhibited expression of the downstream targets of mTOR, such as Raptor and Rictor. The mTOR drives a feedback loop that normally keeps PI3K activity. Overall, fisetin reduces expression of PI3K as well as the level of AKT and mTOR phosphorylation.

Despite the promising results, fisetin, as an anti-cancer agent, regulates multiple signaling pathways that involved in various cellular processes and functions related to cancer development. Therefore, future research should carefully elucidate the anticancer mechanism of fisetin and its impact on these pathways.

In addition, future research should focus on developing advanced delivery systems to address the limitation of bioactive compounds in cancer therapy such as poor solubility, limited bioavailability, toxicity, and off-target effects. This would enhance the therapeutic efficacy, minimize side effects, and achieve targeted therapeutic outcomes (Zare et al., 2021). For example, encapsulated fisetin demonstrated a markedly high therapeutic effect compared to its free form. Fisetin micelles increased cytotoxicity and inhibition of tumor growth (Xiao et al., 2018). We propose the utilization of conjugated micelles as delivery system for fisetin to improve its therapeutic efficacy and minimize toxicity (Pawar et al., 2018). Fisetin-encapsulated enhanced anti-cancer activity and targetability (Aboushanab et al., 2023; Chen et al., 2015b). Hence, further studies are recommended to explore innovative nanocarrier delivery strategies including, niosomes.

Conclusions

The present study demonstrates the anticancer properties of fisetin. Fisetin significantly inhibited cell proliferation in a concentration- and time-dependent manner, demonstrating a significant cell growth inhibition in the Caco-2 cell line. In addition, fisetin significantly modulated the level of cancer-related genes such as those involved in the PI3K/AKT/mTOR pathway, apoptotic genes, BAX and BCL-2, and the transcription factor gene NF-κB. There was a significant increase in BAX, and a significant decrease in BCL-2 expression was observed in response to fisetin in Caco-2 cells. Likewise, the expression of PI3K, AKT, mTOR and NF-κB was significantly decreased in fisetin treated Caco-2 cells. In conclusion, the present study demonstrates the anticancer effects of fisetin and these effects appear to be mediated through the downregulation of the cell survival pathway and the induction of apoptosis.

Supplemental Information

Supplemental Information 1 MTT assay data

Values represent the absorbance of each sample measured at 405nm using BioTek ELx800 absorbance microplate reader

Supplemental Information 2 qPCR data showing ct value for each samples

Fold changes for each gene were calculated using the comparative threshold cycle method (2−ΔΔCt) after normalizing to the values of the GAPDH housekeeping gene and relative control samples

Supplemental Information 3 MIQE checklist

Additional Information and Declarations

Competing Interests

Author Contributions

Data Availability

The authors declare there are no competing interests.

Amal Alamoudi performed the experiments, prepared figures and/or tables, and approved the final draft.

Khlood Alqarni performed the experiments, prepared figures and/or tables, and approved the final draft.

Arwa Ishaq A. Khayyat performed the experiments, analyzed the data, authored or reviewed drafts of the article, and approved the final draft.

Tajamul Hussain conceived and designed the experiments, analyzed the data, authored or reviewed drafts of the article, and approved the final draft.

Salman Alamery conceived and designed the experiments, analyzed the data, prepared figures and/or tables, financial funding, and approved the final draft.

The following information was supplied regarding data availability:

The absorbance and CT values for the experiment are available in the Supplemental Files.

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
