# Peer review of "Modulation of PI3K/AKT/mTOR and apoptosis pathway in colon cancer cells by the plant flavonoid fisetin"

_PeerJ, doi:10.7717/peerj.20225_

## Round 0.1 · original submission · Major Revisions

Please address the concerns of both reviewers and amend the manuscript accordingly.

**Language Note:** The review process has identified that the English language must be improved. PeerJ can provide language editing services - please contact us at [email protected] for pricing (be sure to provide your manuscript number and title). Alternatively, you should make your own arrangements to improve the language quality and provide details in your response letter. – PeerJ Staff

Reviewer 1 ·

Basic reporting

This study examines the anticancer properties of fisetin, a flavonoid derived from plants, on Caco-2 colon cancer cells, emphasizing its impact on the PI3K/AKT/mTOR and apoptosis pathways. Caco-2 cells received treatments with various fisetin concentrations (15–120 μM) over 12 or 24 hours. Cell viability was evaluated using the MTT assay, while gene expression levels of PI3K, AKT, mTOR, NF-κB, BAX, and BCL-2 were analyzed through qRT-PCR. Fisetin markedly decreased cell viability in a dose- and time-dependent manner, leading to the downregulation of PI3K, mTOR, NF-κB, and BCL-2, alongside the upregulation of BAX, which suggests the induction of apoptosis and the suppression of cell survival pathways. Analyses from GeneMANIA and OncoDB further corroborated these results, underscoring fisetin’s potential as a preventative measure against cancer. These findings enhance fisetin’s therapeutic prospects for colorectal cancer, although significant revisions are required prior to publication.

Comments for authors
Comment 1: Revise the abstract to clearly specify the concentrations of fisetin used in the study. This will enhance transparency and allow readers to better understand the experimental design and outcomes.

Comment 2. Expand the introduction to include a detailed discussion of recent advancements in natural compounds targeting colorectal cancer pathways. This will provide a more comprehensive scientific context and strengthen the rationale for the study.

Comment 3. Confirm the provenance of the Caco-2 cell line used in the experiments. Include authentication details to ensure the reliability and reproducibility of the results.

Comment 4. Explain why AKT mRNA levels did not show significant changes despite its central role in the PI3K/AKT/mTOR pathway. Clarifying this observation is essential to understanding the modulation of the pathway by fisetin.

Comment 5: Discuss the potential off-target effects of fisetin on other signaling pathways. Providing this analysis will offer a more balanced and critical perspective on the compound’s specificity and therapeutic potential.

Comment 6: Address whether fisetin’s effects on the PI3K/AKT/mTOR pathway are direct or mediated through upstream regulators. Clarify the mechanism of action to enhance the scientific depth of the study.

Comment 7. Correct inconsistent terminology throughout the manuscript. Carefully proofread the text to fix grammatical errors, improve sentence flow, and standardize terminology to ensure clarity and coherence.

Experimental design

-

Validity of the findings

-

·

Basic reporting

I doubt the novelty of this work.

Experimental design

It's Ok.

Validity of the findings

As expected, flavonoids are well known to inhibit cancer growth.

Additional comments

Dear authors,

Clearly highlight the novelty of your work in both the abstract and introduction sections. Since flavonoids are already well known for their inhibitory effects on colorectal cancer (CRC), it is essential to emphasize what distinguishes your study from existing literature.

In the discussion section, cite recent and relevant references to strengthen the scientific context and demonstrate engagement with the latest research developments.

Include a forward-looking perspective on your work within the methodology section. Specifically, discuss the potential for encapsulating the studied molecule in delivery systems such as niosomes to achieve targeted therapeutic effects. Explain how this approach could enhance specificity and clinical applicability.

Add a schematic representation that illustrates the proposed mechanism of action and the central hypothesis of your study. This visual summary will help clarify your research framework and improve the manuscript’s overall impact.

---

## Round 0.2 · accepted · Accept

Issues pointed out by the reviewers are addressed and the revised manuscript is acceptable now.

Reviewer 1 ·

Basic reporting

The revised version is improved and deserved to be published in present form.

Experimental design

The revised version is improved and deserved to be published in present form

Validity of the findings

The revised version is improved and deserved to be published in present form

Additional comments

The revised version is improved and deserved to be published in present form